# Inactivated Avian Infectious Bronchitis Virus Strains M41 and 4–91 Provide Broad Protection Against Multiple Avian Infectious Bronchitis Strains

**DOI:** 10.3390/vaccines14010039

**Published:** 2025-12-29

**Authors:** Noortje M. P. van de Weem, Mateusz Walczak, Lieke van Rooij, Frank A. J. Hormes, Peter Hesseling, Lieke Timmers, Pieter A. W. M. Wouters, Rüdiger Raue

**Affiliations:** MSD Animal Health, 5831 AN Boxmeer, The Netherlands; noortje.weem@msd.com (N.M.P.v.d.W.);

**Keywords:** Infectious Bronchitis Virus, IB, IBV, inactivated poultry vaccine, cross-protection, M41, 4–91, QX, Q1, Var2

## Abstract

**Background/Objective**: The poultry industry requires extensive vaccination of chickens against IBV in an effort to prevent the disease in animals and significant economic losses. Current vaccination strategies often lack effectiveness, and the continual emergence of new IBV variants makes disease control increasingly challenging. We have developed an inactivated vaccine for poultry containing nine different antigens (Nobilis Multriva), including two IBDV strains, two ARV strains, one NDV strain, one AMPV strain, one EDSV strain and two IBV strains: M41 (genotype GI-1) and 4–91 (genotype GI-13). In this study, the IB efficacy of this novel inactivated vaccine was investigated against homologous and heterologous IBV strains. **Methods**: Inactivated IBV vaccine containing the M41 and 4–91 strains (Nobilis Multriva) was administered intramuscularly, either alone or following vaccine priming, in SPF and commercial chickens. Birds were challenged with homologous and heterologous IBV strains at defined ages (peak of lay, mid-lay and end of lay). Vaccine efficacy was evaluated through serological assays, clinical observations, and monitoring of egg production post-challenge. **Results**: This vaccine provided excellent broad protection against different IBV strains circulating in different parts of the world, including IBV M41, 4–91, QX, Q1 and Var2. Furthermore, the vaccine provided long-lasting IBV serological response against IB M41 and IB 4–91 until at least 96 weeks of age in SPF and commercial layers and breeder birds. This vaccine will allow farmers to reduce the number of vaccination moments, thereby minimizing stress to the birds, while also decreasing labor demands and the risk of human error, ultimately contributing to lower overall vaccination costs. **Conclusions**: Given its demonstrated broad cross-protection and sustained serological responses, this nine-valent inactivated vaccine (Nobilis Multriva) represents a key component of an effective vaccination regimen for controlling IBV infections in the poultry industry.

## 1. Introduction

Avian Infectious Bronchitis (IB) is a highly contagious poultry disease, affecting the global poultry population and causing significant economic losses to the poultry industry [1]. IB is caused by Infectious Bronchitis Virus (IBV) entering via the respiratory tract. The virus replicates in the epithelial cells of respiratory tract, kidney and reproductive organs [2,3]. Clinical signs usually develop within 24 to 48 h after infection, and virus concentrations in the trachea peak during the first 3 to 5 days. Young chicks exhibit gasping, coughing, nasal discharge, sneezing, watery eyes and lethargy; some birds may die directly from IBV [3]. Furthermore, IBV-induced lesions in the respiratory tract often increase susceptibility to secondary respiratory infections [2,3]. In layers and breeders, IBV infections often contribute to diminished egg production and quality, and in broilers, it may result in growth retardation [2,3]. IBV spreads by aerosols and feces [2,3].

IBV is an enveloped RNA gammacoronavirus belonging to the *Coronaviridae* family [2]. The IB virion consists of the structural spike (S) subunit 1 and subunit 2, envelope (E), membrane (M) and nucleocapsid (N) proteins. The S1 protein is involved in host cell attachment and contains virus-neutralizing epitopes. Recombination in the S glycoprotein gene can result in the emergence of new strains. Large numbers of IBV variants circulate in the world, of which some are unique to a local area while others occur at many locations [4,5]. IBV contains at least six genotypes with a total of 32 distinct viral lineages and a few unassigned inter-lineage recombinants [6]. A recent review paper describes that there may be nine genotypes of IBV [7]. This paper emphasizes that Infectious Bronchitis Virus (IBV) exhibits remarkable genetic diversity, with distinct genotypes distributed across different regions worldwide. Genotype I (GI) is globally prevalent, while GII primarily originated in Europe and later spread to South America. Other lineages show regional confinement, such as GIII and GV in Australia, GVII in Europe and China, GVI in Asia and GIV, GVIII and GIX in North America, highlighting the complex epidemiology of IBV [7].

In order to visualize this diversity of IBV strains, we have prepared an IBV phylogenetic tree based on genetic characterization of the S1 gene, which illustrates the broad variety of IBV strains (Figure 1).

As IBV is an RNA virus characterized by high mutation and recombination capacity [5,8]; new IBV strains keep emerging and challenging the poultry industry. Commercially available IBV vaccines include live attenuated and inactivated types, but their effectiveness is often compromised by the genetic diversity of circulating field strains. These challenges have prompted the development of improved vaccination strategies. Currently, live attenuated vaccines are used in young birds, or for priming of future layers and breeders that will be boosted with inactivated vaccines before start of lay [4,9]. Current monovalent IB vaccines (both live attenuated and inactivated) tend to provide limited protection against new emerging heterologous strains [4,9,10]. Therefore, effective (inactivated) IB vaccines and prime–boost regimens are needed that will allow broader protection for layers and breeder birds [1].

We have developed an inactivated vaccine for poultry (Nobilis Multriva) containing nine different antigens, including two IBDV strains (GB02 and 89/03), two ARV strains (ARV-1 and ARV-4), one NDV strain (Ulster), one AMPV strain (BUT1 #8544), one EDSV strain (BC14) and two globally prevalent IBV strains (M41; genotype GI-1 and 4–91; genotype GI-13). The 4–91 strain has been reported in Canada, Brazil, Europe, Japan, China, other parts of Asia, Russia, Africa and Pakistan, whereas the M41 (Mass-type) strain exhibits a global distribution, including the USA, Canada, South America, Asia, Pakistan and Europe [11]. In this study, the IB efficacy of this inactivated vaccine was investigated against homologous IBV M41 and IBV 4–91 strains, as well as heterologous strains including Q1 (genotype GI-16), QX (genotype GI-19) and Var2 (genotype GI-23), which were selected to represent genetically and antigenically diverse field-relevant IBV variants circulating in poultry populations (Figure 1). The efficacy was assessed in both specific pathogen-free (SPF) and different types of commercial field birds.

## 2. Materials and Methods

### 2.1. Vaccines

The following vaccines were used for IBV live prime vaccinations: Nobilis IB Ma5 (MSD Animal Health), Nobilis IB 4–91 (MSD Animal Health), Nobilis IB Primo QX (MSD Animal Health) and/or Poulvac IB Primer (containing IB H120 & IB D274; Zoetis; Girona, Spain). Vaccines were administered at the manufacturer’s recommended dose by the recommended route.

In all animal studies, an inactivated IBV vaccine formulated as a water-in-oil emulsion was administered either as a single vaccination or as a booster. The vaccine used was Nobilis Multriva RT+IBm+ND+Gm+REOm+EDS (MSD Animal Health) or Nobilis Multriva RT+IBm+ND+Gm (MSD Animal Health).Nobilis Multriva contains nine different antigens: two IBDV strains (GB02 and 89/03), two ARV strains (ARV-1 and ARV-4), one NDV strain (Ulster), one AMPV strain (BUT1 #8544), one EDSV strain (BC14) and two IBV strains (M41; genotype GI-1 and 4–91; genotype GI-13). The inactivated vaccines were administered via the intramuscular (IM) route at a dose of 0.3 mL/bird.

### 2.2. IBV Challenge Strains

Challenge strains were selected based on the IBV phylogenetic tree to represent homologous and heterologous strains (Figure 1). The following IBV challenge strains were used: M41 (genotype GI-1), 4–91 (genotype GI-13), Q1 (genotype GI-16), QX (genotype GI-19) and Var2 (genotype GI-23). Each challenge strain was reconstituted and diluted in an amount of cold sterile demineralized water to achieve a titer from 6.0 to 7.1 log10 EID50/bird. The reconstituted challenge strains were stored on melting ice until application via the intratracheal route, or for the IB M41 challenge at 26 weeks of age (woa) via the intraocular route.

### 2.3. Experimental Procedures

An overview of the experimental designs of the various studies is given in Table 1. A total of 20 to 48 birds were included per group in all animal studies (Table 2).

### 2.4. Studies in SPF White Leghorn Layers

SPF white Leghorn layers were prime-vaccinated with Nobilis IB Ma5 and Nobilis IB 4–91 and boosted with Nobilis Multriva RT+IBm+ND+Gm+REOm+EDS or Nobilis Multriva RT+IBm+ND+EDS (Table 1). Some of the animals were only vaccinated with Nobilis Multriva. At 1, 4, 8, 12, 15, 20, 24, 30, 40 49, 60, 70, 80 and 90 weeks of age (woa), blood samples were collected individually from the birds for serological evaluation.

At different timepoints after vaccination, the efficacy of the vaccines was examined by challenging laying hens with different IBV strains. The IB M41 and IB 4–91 efficacy was examined at onset of immunity (26–27 woa), in the middle of the hen lifecycle (60 woa) and at the end of the hen lifecycle (96 woa) by monitoring egg production after challenge. The heterologous efficacy was tested at 26 woa for Q1, at 37 woa for Var2 and at 40 woa for QX, by challenging birds with one of those strains. The baseline for the egg production pre-challenge was established as the average daily egg production in the 7 days preceding challenge.

### 2.5. Studies in Commercial Field Chickens

Field chickens (layers: Lohman Brown Lite or H&N Super Nick; broiler breeders: Ross 308) were primed with live Nobilis IB Ma5, Nobilis IB 4–91, Poulvac IB primer and/or Nobilis IB Primo QX and boosted with Nobilis Multriva (Table 1). Starting from approx. 10 woa, blood samples were collected from birds at intervals of approximately 10 weeks for serological evaluation.

### 2.6. Clinical Observations

Throughout the experiments, the chickens were observed at least once daily for clinical signs. After IBV challenge, birds were observed for IB-related clinical signs. Coughing/snicking was one of the observed parameters, and the occurrence was recorded per pen. A score of 1 was given when coughing was recorded in a pen; a score of 2 was given when multiple birds were coughing in a particular pen. For each group, the total score was calculated by adding all individual daily scores recorded in all pens of a particular group.

### 2.7. Haemagglutination Inhibition (HI) Assay

Blood samples collected from birds were centrifuged, and obtained sera were heat-inactivated (30 min at 56 °C). Those sera samples were analyzed for the presence of haemagglutination-inhibiting antibodies using IBV M41-specific or IBV 4–91-specific haemagglutination inhibition assay [12]. The IBV antigens were prepared from allantoic fluid harvested from IBV-inoculated embryonated SPF eggs and concentrated 100-fold by ultracentrifugation (54,000× *g*, 60 min, 4 °C). The concentrated IBV strains were treated with neuraminidase (for 3 h at 37 °C) and were used as HA antigens in the corresponding HI test [13]. In brief, for the HI test, samples were diluted in a 2-fold serial dilution and mixed with an equal volume of IBV M41 or IBV 4–91 antigen (4–8 HAU). After 45 min incubation at 2–8 °C, 1% chicken red blood cells were added, and after a second incubation of 1 h at 2–8 °C, the dilutions were examined for inhibition of the haemagglutination. The HI titer is expressed as log2 of the reciprocal of the highest serum dilution that completely inhibits hemagglutination.

### 2.8. Statistical Analysis

Serological group comparison was performed using the exact Wilcoxon rank-sum test.

Egg production rates were statistically modeled based on a generalized linear model using the log link function and the Poisson distribution to model the weekly egg numbers per pen and using the logarithm of the number of hen-days in the respective week of the pen as an offset to analyze the egg laying rates. GEE estimation in combination with the empirical standard error dealt with the correlation in the repeated measurements in the pens over the four weeks after challenge. In order to compare post-challenge egg production to baseline production, a separate analysis was performed per vaccination group. The model included week-after-challenge as a factor, and Least-Squares-Means (LSMs) comparisons were performed for the results of each week after challenge to the production at the pre-challenge week (week 0). The vaccination group comparison was performed using the model with factors: group, week-after challenge and the interaction of these two factors. Where the interaction was not statistically significant, an interpretation of the direct group comparison of the LSMs for the group was provided.

### 2.9. Ethical Statement

All experiments were conducted in accordance with the Animal Health and Welfare regulations and approved by the Animal Welfare Body of the Royal GD and MSD Animal Health and were registered according to the Dutch legislation. Experimental procedures were conducted in an ethical and responsible manner, adhering to all codes of experimentation and legislation. Field trial studies were done in conformity with Good Clinical Practice.

## 3. Results

### 3.1. Vaccination with Nine-Valent Inactivated Nobilis Multriva Vaccine Induces Efficient and Long-Lasting IBV Serological Responses (Against IBV M41 and IBV 4–91) in SPF Chickens Alone or in a Prime–Boost Regimen

SPF layers were given an inactivated RT+IBm+ND+Gm+REOm+EDS vaccination (at 15 woa) alone or in combination with an IB live prime vaccination (Nobilis IB M41 and Nobilis IB 4–91; at 1 doa). The IBV M41 and IBV 4–91 specific antibody titers were measured using the HI test at 1, 4, 8, 12, 15, 20, 24, 30, 40, 49, 60, 70, 80 and 90 woa and are displayed in Figure 2a,b.

Vaccination with only the inactivated (Multriva) vaccine resulted in an average IBV-specific antibody titer of 7.5 log2 (IBV M41) and 7.9 log2 (IBV 4–91) at 20 woa (5 weeks post-vaccination). Live prime vaccination with Nobilis IB M41, Nobilis 4–91, followed by a boost with the inactivated (Multriva) vaccine, increased theses IBV-specific antibody titers to 9.7 log2 (IBV M41) and 11.4 log2 (IBV 4–91).

Similar serological profiles, for both IBV M41 and IBV 4–91 HI antibodies, were observed for birds vaccinated with the inactivated vaccine. Titers remained at baseline levels before vaccination, increased to a maximum by 4 weeks post-vaccination (wpv) and persisted at high levels up to 90 woa. The IBV M41 titers were between 6.1 and 7.1 log2 and the IBV 4–91 titers between 7.2 and 8.7 log2 in the time period between 30 and 90 woa. The prime–boost vaccinated birds showed a live prime vaccination response from 8 to 12 woa, with titers increasing following the inactivated booster vaccination. Titers persisted between 8.2 and 9.7 log2 for IBV M41 and between 10.1 and 11.4 log2 for IBV 4–91 for up to 90 woa. Non-vaccinated birds showed background levels of non-specific reactivity for both IBV M41 antibodies and IBV 4–91 antibodies typical of the age of the birds (≤5 log2).

This data shows that vaccination with nine-valent inactivated Nobilis Multriva vaccine, alone or in a prime–boost regimen, induces stable and long-lasting antibodies in SPF birds against IBV M41 and IBV 4–91 up to at least 90 woa.

### 3.2. Vaccination with Nine-Valent Inactivated Nobilis Multriva Vacine Induces Efficient and Long-Lasting IB Serological Responses (Against IBV M41 and IBV 4–91) Under the Field Conditions in Commercial Chickens

Several different commercial breeds of chickens (layers: Lohman Brown Lite and H&N Super Nick, broiler breeders: Ross 308) were vaccinated with the nine-valent inactivated Nobilis Multriva vaccine between 12–15 woa as part of field vaccination protocols. Layers were primed with Nobilis IB Ma5, Nobilis IB 4–91 and Poulvac IB primer or Nobilis IB Primo QX, and broiler breeders with Nobilis IB Primo QX, Nobilis IB Ma5 and Nobilis IB 4–91 (Table 1).

Across all chicken breeds, HI serological responses to both IBV M41 and IBV 4–91 remained high after booster vaccination with the inactivated vaccine, ranging from 7.4 to 12.0 log2 for IBV M41 and from 10.4 to 12.1 log2 for IBV 4–91, and persisted to at least 100 weeks of age (woa) (Figure 3a,b). The levels of IBV-specific antibodies were comparable between different breeds of commercial chickens.

This indicates that vaccination with nine-valent inactivated Nobilis Multriva vaccine, under field conditions, induces stable and long-lasting antibodies against IBV M41 and IBV 4–91 in different breeds of commercial chickens, both layers and broiler breeders.

### 3.3. Birds Vaccinated with Nine-Valent Inactivated Nobilis Multriva Vaccine Are Effectively Protected Against IBV Homologous Challenges

SPF layers were given an inactivated vaccination alone (15–16 woa) or in combination with an IB live prime vaccination (Nobilis IB M41 and Nobilis IB 4–91; 1 doa). The egg-lay percentage of these birds was determined before and after challenge with IBV M41 or IBV 4–91 (at 26 woa, 60 woa and 96 woa to determine homologous IBV protection.

Prior to challenge IBV M41 and IBV 4–91 HI antibody titers were determined. Birds vaccinated only with the nine-valent inactivated Nobilis Multriva vaccine or with a prime–boost vaccination schedule had significantly higher mean antibody titers (*p* < 0.001; exact Wilcoxon rank-sum test) compared to the non-vaccinated birds (Table 2). All birds in the non-vaccinated group showed levels of non-specific reactivity (≤5 log2 HI) for both IBV M41 and IBV 4–91, which is typical for birds of this age.

The baseline egg production before IBV M41 challenge at 26 woa was approximately 92% (Figure 4a). In the non-vaccinated group, the egg production after challenge dropped to 81% in week 2 (*p* < 0.0106) of the pre-challenge baseline. The egg production of the group vaccinated only with inactivated vaccine was slightly lower (3%; *p* = 0.0237) as compared to the pre-challenge baseline. The group vaccinated with both prime and inactivated boost vaccine did not show any drop in egg production compared to the pre-challenge baseline. The baseline egg production around 60 woa was approximately 78% (Figure 4c). The egg production of the non-vaccinated group decreased to 39% (*p* = 0.0045) in two weeks after IBV M41 challenge (Figure 4c). The vaccinated prime + boost birds showed slightly reduced egg production from weeks 1 to 3, of which week 3 was different compared to the baseline (*p* < 0.0001), but even then remained high, at around 93.4% of the production in week 0 before challenge (Figure 4c). At 96 weeks of age (Figure 4e), non-vaccinated birds showed up to a 50% reduction in egg production in week 2 after challenge compared with the pre-challenge baseline (*p* < 0.0001). The prime–boost vaccinated group showed only a slight reduction at 1 week post-challenge, dropping to 87% (*p* = 0.0287) compared to the pre-challenge baseline.

Birds vaccinated with the nine-valent inactivated Nobilis Multriva vaccine, alone or in combination with a live prime, maintained a high egg production rate after IBV M41 challenge in comparison to the non-vaccinated birds, which experienced significant reduction in egg production.

The baseline egg production before IBV 4–91 challenge at 26 woa was approximately 95.3% (Figure 4b). In the non-vaccinated group, the egg production after IBV 4–91 challenge dropped to 86% in week 2 (*p* < 0.0427) of the pre-challenge baseline. The egg production of the groups vaccinated only with the nine-valent inactivated Nobilis Multriva vaccine, or vaccinated with both live prime and inactivated boost vaccine, was reduced by less than 5% compared to the pre-challenge baseline. The baseline egg production at around 60 woa was approximately 73% (Figure 4b). The egg production of the non-vaccinated group decreased to 76% (*p* < 0.0001) by two weeks post-challenge (wpc). The egg production in the vaccinated prime + boost birds remained at normal pre-challenge levels. At 96 woa (Figure 4f), a reduction in egg production was observed for the non-vaccinated birds up to 58% (*p* < 0.0001) by 2 wpc. Egg production in the prime–boost vaccinated group remained at pre-challenge levels throughout the monitoring period.

In summary, birds vaccinated with the nine-valent inactivated Nobilis Multriva vaccine, alone or in combination with a live prime, maintained a high egg production rate after IBV 4–91 challenge, while unvaccinated birds had a significant reduction in egg production. Overall, a live prime–boost vaccination provides protection against homologous IBV M41 or IBV 4–91 challenge to at least 96 woa.

### 3.4. Birds Vaccinated with Nine-Valent Inactivated Nobilis Multriva Vaccine Are Effectively Protected Against IBV Heterologous Challenges

Specific-pathogen-free (SPF) layers received either a single inactivated IB vaccination at 15–16 weeks of age or a combined regimen consisting of an IB live prime (Nobilis IB M41 and Nobilis IB 4–91 at 1 day of age) followed by the inactivated vaccine at 15–16 weeks. Egg production was monitored before and after challenge with IBV Q1 (26 weeks), IBV Var2 (37 weeks) or IBV QX (40 weeks) to assess heterologous IBV protection.

The baseline egg production before IBV QX challenge at 40 woa was approximately 90% (Figure 5a). The egg production of the non-vaccinated group was reduced to 49.4% (*p* < 0.0001) by week 2 after challenge compared to the pre-challenge baseline (Figure 5a). The egg production of the vaccinated group was slightly lower, with 90.7% (*p* = 0.0418) at 2 wpc compared to the pre-challenge baseline.

The baseline egg production before IBV Var2 challenge at 37 woa was approximately 92% (Figure 5b). Egg production in the non-vaccinated group was significantly reduced to 63.9% by two weeks post-challenge compared with the pre-challenge baseline (*p* < 0.0001; Figure 5b). The egg production of the vaccinated group remained very high, but was significantly lower with 91.3% (*p* = 0.0001) at 2 wpc compared to the pre-challenge baseline.

The baseline egg production, before IBV Q1 challenge at 26 woa, was approximately 96% (Figure 5c). The egg production of the non-vaccinated group was significantly reduced to 60% (*p* < 0.05) by two weeks post-challenge compared to the pre-challenge baseline (Figure 5c). The egg production of the vaccinated group remained high at >91%.

This data shows that birds vaccinated with a live IBV prime in combination with the nine-valent inactivated Nobilis Multriva vaccine exhibited markedly improved protection against heterologous IBV strains QX, Var2 and Q1, as evidenced by a higher percentage of egg production compared to non-vaccinated controls.

### 3.5. Vaccination with Nine-Valent Inactivated Nobilis Multriva Vaccine in a Prime–Boost Regimen Reduces Respiratory Signs Induced by Different IBV Strains

For each IBV challenge, the occurrence of coughing was recorded on a daily basis and summarized per group (Table 3). In general, coughing was observed within the first two wpc with high frequencies in the first few days post-challenge.

After IBV M41 challenge at 26 woa, substantial coughing was observed in the non-vaccinated control groups. In contrast, birds vaccinated with the nine-valent inactivated Nobilis Multriva vaccine, alone or in a prime–boost schedule, had minimal coughing. Challenge at 60 and 96 woa resulted in observations of coughing within the first week for the non-vaccinated birds and no observations for the vaccinated birds. IBV 4–91 challenge resulted in similar observations as for IBV M41 even though some birds receiving only inactivated vaccine were also coughing. Following IBV QX challenge, both non-vaccinated and IBV prime–boosted birds presented with coughing. Coughing was however less severe and had a shorter duration in the vaccinated birds. After IBV Var2 challenge, substantial coughing (score 34) was recorded for the non-vaccinated birds. In contrast, coughing was nearly absent (score 2) in the IBV prime–boosted birds.

Overall, it can be seen that vaccination with nine-valent inactivated Nobilis Multriva vaccine in a prime-boost regimen is efficacious in reducing respiratory signs (coughing) following infection with different homologous and heterologous IBV strains.

## 4. Discussion

We have developed an inactivated vaccine for poultry (Nobilis Multriva) containing nine different antigens, including two IBDV strains (GB02 and 89/03), two ARV strains (ARV-1 and ARV-4), one NDV strain (Ulster), one AMPV strain (BUT1 #8544), one EDSV strain (BC14) and two globally prevalent IBV strains (M41; genotype GI-1 and 4–91; genotype GI-13). In this study, we investigated the efficacy of this newly developed inactivated vaccine against different homologous and heterologous IBV strains. This inactivated vaccine was able to induce long-lasting IBV serological responses in both SPF and commercial chickens. Chickens vaccinated with this vaccine were effectively protected against homologous IBV challenges up to at least 96 weeks of age. Furthermore, the vaccine provided protection against important heterologous IBV strains such as Var2, Q1 and QX, showcasing the broad IBV coverage. Hens vaccinated with the nine-valent inactivated Nobilis Multriva vaccine were protected against drops in egg production and respiratory signs (e.g., coughing) caused by IBV infection. The breadth of protection was confirmed through serology studies under both controlled and field conditions, indicating robust efficacy across diverse genetic backgrounds. Importantly, these combined outcomes suggest that the vaccine not only mitigates clinical disease but may play an important strategic role in reducing economic losses associated with IBV outbreaks.

All type of commercial birds, especially layers and breeders, are extensively vaccinated against IBV to prevent the disease and minimize economic losses to the global poultry industry. Vaccine effectiveness depends on multiple aspects, from which optimal antigen combination is of most importance [1,14]. The emergence of new strains continues to pose challenges for effective IB control, and current IBV vaccination strategies may not always provide optimal protection [15]. Development of IB vaccines providing broad protection is therefore necessary [4]. Vaccination with multiple live attenuated genotypes (e.g., double IBV Massachusetts-type (Mass), IBV BR1 with Mass or IBV Mass with IBV 4–91) could provide cross-protection against heterologous virus strains [14,16,17,18], but most IBV vaccine strains induce poor cross-protection [3,5,19,20]. It is suggested that vaccines containing a single IBV strain with a low level of homology in the S1 gene with an infecting IB strain are less efficient in inducting good cross-protection [4]. On the other hand, it has been demonstrated that combination of two distinct live vaccines such as IB Ma5 and 4–91 could provide broad protection against respiratory infections caused by D1466, ARK [21] and QX [17] at young age. Only a few studies showed that vaccination with double live prime (IB Mass or Ma5 with IB 4–91) in combination with an inactivated IB vaccine (M41 or Mass with BR-1 or Mass with D274) decreased the egg-drop percentage against M41, 4–91, QX, Q1, Var2 challenge [22,23]. The effect of a prime vaccination in combination with inactivated IBV M41 and IBV 4–91 has not yet been demonstrated. Our study demonstrates the excellent cross-protection of a nine-valent inactivated Nobilis Multriva vaccine containing IB strains M41 and 4–91, given in combination with prime, against different circulating IBV strains, including M41, 4–91, QX, Q1 and Var2. The protection was demonstrated around the time of peak egg production at 27 weeks. Moreover, the occurrence of respiratory signs was also reduced with this vaccination regimen.

The mechanisms underlying why certain IBV strains confer broader cross-protection while others fail to do so remain unclear [10]. Differences in genetic background of bird breeds could play a role in the susceptibility to IBV [24]. To investigate this aspect, we also examined the effectiveness of the inactivated multivalent vaccine in inducing anti-IBV serological responses in different commercial birds: Lohmann Brown Lite and H&N Super Nick Layers and in Ross 308 Broiler Breeders. Administration of the nine-valent inactivated Nobilis Multriva vaccine in combination with prime vaccination(s) under field conditions confirmed the long-life serological responses seen in SPF birds and thus the effectiveness of this inactivated vaccine in a prime–boost regimen in commercial birds.

Worldwide, farmers are interested in long and persistent egg lay production and stability in egg quality [25] while keeping hens in production until 100 weeks of age or longer [26]. One of the important steps to reach this goal is to protect extended long-life flocks against infections. In this study, we demonstrated that the nine-valent inactivated Nobilis Multriva vaccine, given in combination with prime vaccination, provides long-lasting IBV serological response against IBV M41 and IBV 4–91 in both SPF and commercial layer and broiler breeder birds. A prolonged serological response against IBV M41 and IBV 4–91 is essential for broiler breeders and layers, as their extended production lifespan requires sustained immunity.

In summary, the multivalent inactivated vaccine containing the IBV strains M41 and 4–91 offers effective protection against a variety of homologous and heterologous IBV strains which circulate in the world. These results confirm and expand the IB protectotype concept [23]. With the broad cross-protection and efficacy, this inactivated vaccine will have a beneficial impact in the poultry industry to control IBV infections.

## 5. Conclusions

The nine-valent inactivated vaccine Nobilis Multriva demonstrated broad and sustained protection against homologous and heterologous IBV strains in both SPF and commercial birds. Its efficacy in prime–boost regimens and ability to maintain long-lasting serological responses underline its strategic value for improving IBV control and reducing economic losses in poultry production.

## Figures and Tables

**Figure 1 vaccines-14-00039-f001:**
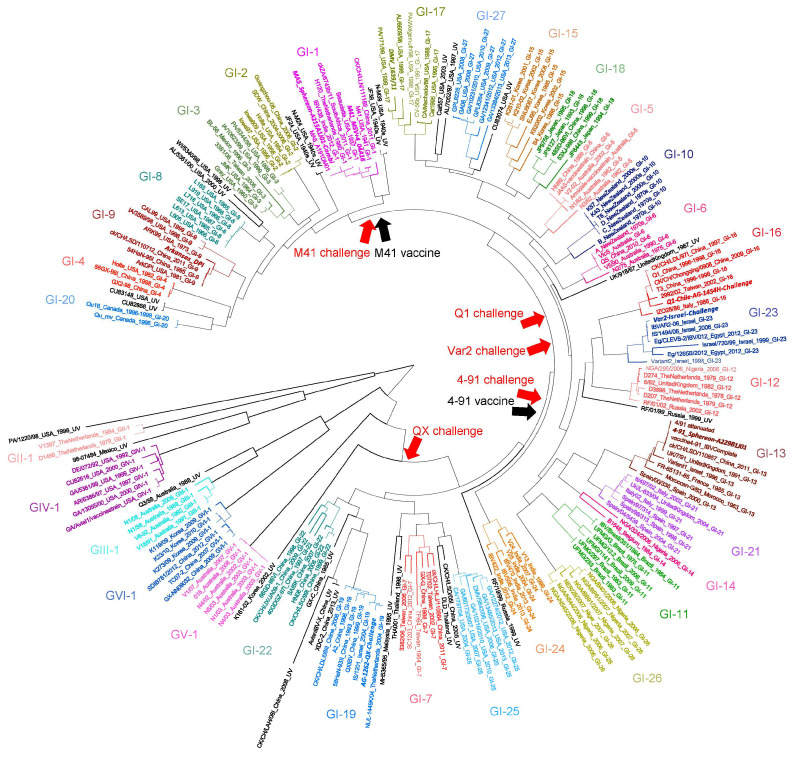
Phylogenetic tree of complete S1 nucleotide sequences of IBV strains. Bars reporting the genotypes in which the lineages (color-coded) fall are shown. Isolate number, country of origin and collection date are given for each strain. The designation “UV” indicates unique variants, marked in black. IBV strains contained in the nine-valent inactivated vaccine are shown together with challenge strains used in this study. The phylogenetic tree was created using MAFFT v7.450 and estimated by RAxML v8.2.11.

**Figure 2 vaccines-14-00039-f002:**
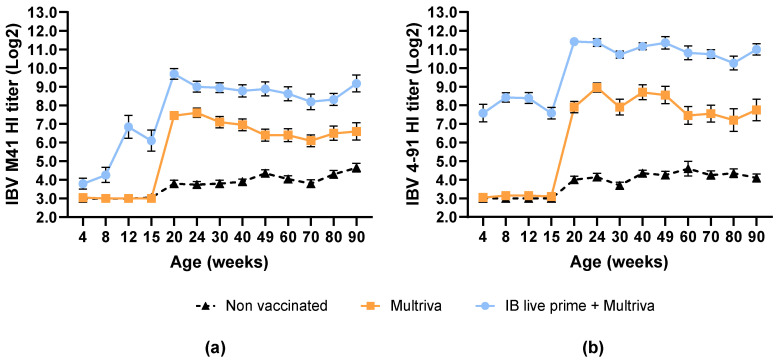
Mean serological responses to IBV strains depicted as (**a**) IBV M41 and (**b**) IBV 4–91 HI titer levels in SPF chickens over time. SPF chickens were given an IBV prime–boost vaccination (blue circles), vaccination with nine-valent inactivated vaccine (Multriva; orange squares) only or no vaccination (dotted triangles). Prime vaccination was done at 1 doa via spray, and inactivated boost vaccination was done at 15 woa using IM route. Each time point is an average of at least 20 animals, and the error bars indicate standard error of the mean.

**Figure 3 vaccines-14-00039-f003:**
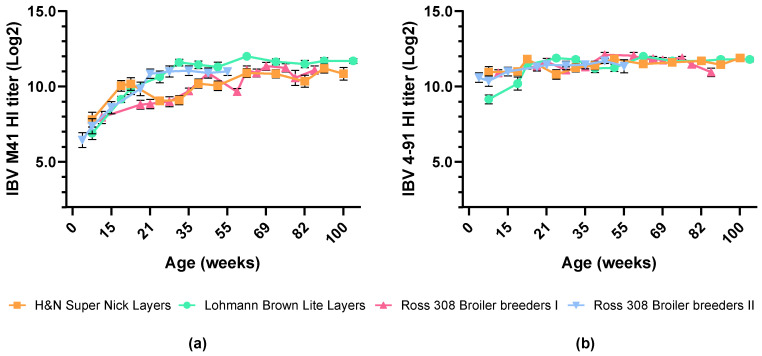
Mean serological responses to IBV strains depicted as (**a**) IBV M41 and (**b**) IBV 4–91 HI titer levels over time in commercial layers (orange squares and green circles) and broiler breeder (pink triangles up and blue triangles down) birds vaccinated with prime–boost regimen under field conditions. Animals were IBV primed at 1 doa and boosted with inactivated vaccine at 12–15 woa. Error bars indicate standard error of the mean.

**Figure 4 vaccines-14-00039-f004:**
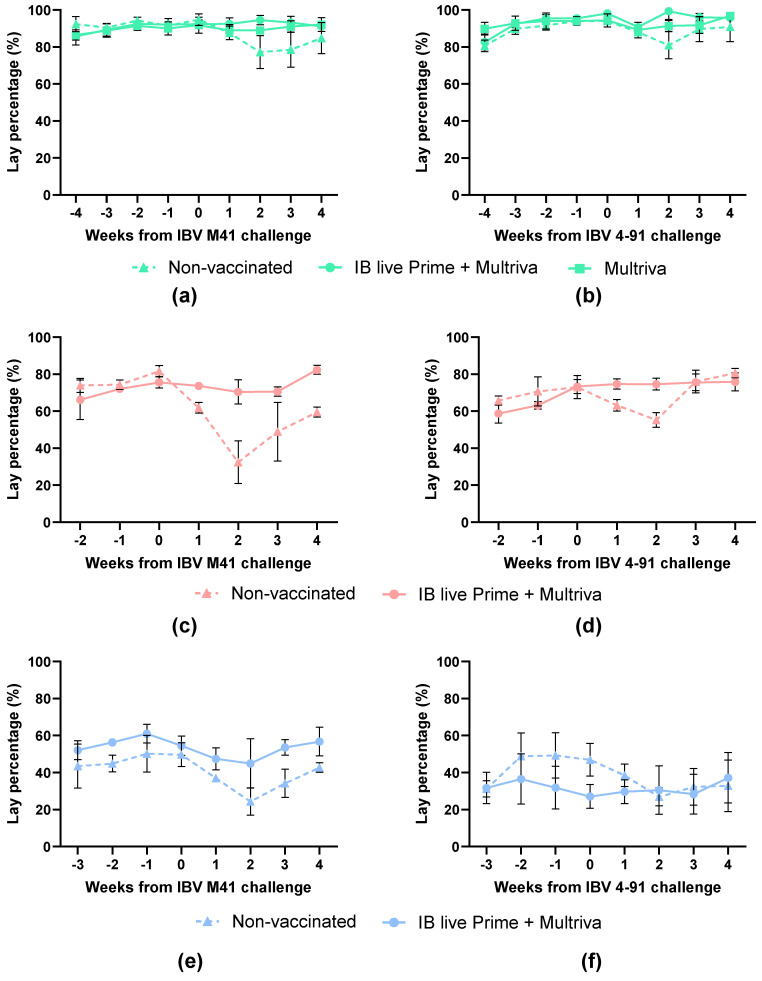
Egg-lay percentage before and after challenge with homologous strains: IBV M41 (**left**) or IBV 4–91 (**right**) at 26 woa (**a**,**b**), at 60 woa (**c**,**d**) and at 96 woa (**e**,**f**). Non-vaccinated birds are indicated with triangles and a dotted line, inactivated vaccine-vaccinated birds (at 15 woa) with squares and prime+boost-vaccinated birds (prime at 1 doa and boost at 15 woa) with circles. Error bars indicate standard error of the mean.

**Figure 5 vaccines-14-00039-f005:**
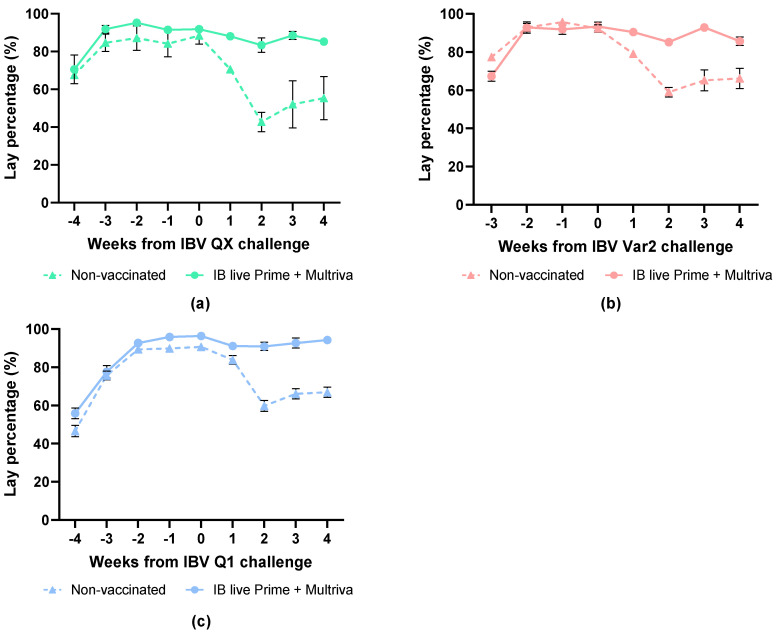
Egg-lay percentage before and after challenge with heterologous strains: IBV QX (**a**), IBV Var2 (**b**) or IBV Q1 (**c**) at 40, 26 and 37 woa, respectively. Non-vaccinated birds are indicated with triangles and dotted lines and prime+boost-vaccinated birds with circles. Error bars indicate standard error of the mean.

**Table 1 vaccines-14-00039-t001:** Overview of studies performed with different bird types which were vaccinated using different IB vaccine prime–boost schedules. Type of vaccine used and age of birds during prime–boost vaccination is also included in the table.

Study Type	Type of Birds	Prime	Boost	Blood Collection	Egg Collection of Birds Challenged at t =
Laboratory studies	SPF White Leghorn Layers	None or Nobilis IB Ma5 (at 1 doa), Nobilis IB 4–91 (at 1 doa)	Nobilis Multriva RT+IBm+ND+Gm+ REOm+EDS (at 15 woa)	1, 4, 8, 12, 15, 20, 24, 30, 40 49, 60, 70, 80, 90 weeks of age (woa)	IB M41: 26, 60, 96 woa IB 4–91: 27, 60, 96 woa QX: 40 woa Var2: 37 woa
		Nobilis IB Ma5 (at 1 doa), Nobilis IB 4–91 (at 1 doa)	Nobilis Multriva RT+IBm+ND+EDS (at 16 woa)	16 and 20 woa	Q1: 26 woa
Field studies	Lohmann Brown Lite Layers	Nobilis IB Ma5 (at 1 doa), Nobilis IB 4–91 (at 1 doa and 2 woa), Poulvac IB Primer (at 10 woa)	Nobilis Multriva RT+IBm+ND+Gm+ REOm+EDS (at 12 woa)	Starting from approx. 10 woa with approx. 10-week intervals	Not applicable
	H&N Super Nick Layers	Nobilis IB Ma5 (at 1 doa), Nobilis IB 4–91 (at 1 doa and 2 woa), Nobilis IB Primo QX (at 5 woa)	Nobilis Multriva RT+IBm+ND+Gm+ REOm+EDS (at 12 woa)		
	Ross 308 Broiler Breeder	Nobilis IB Ma5 (at 1 doa), Nobilis IB 4–91 (at 10 doa), Nobilis IB Primo QX (at 1 doa) Poulvac IB QX (at 9 woa)	Nobilis Multriva RT+IBm+ND+Gm+ REOm+EDS (at 14 woa)		
	Ross 308 Broiler Breeder	Nobilis IB Ma5 (at 1 doa), Nobilis IB 4–91 (at 10 doa), Nobilis IB Primo QX (at 1 doa and 9 woa)	Nobilis Multriva RT+IBm+ND+Gm+ REOm+EDS (at 15 woa)		

doa—days of age, woa—weeks of age.

**Table 2 vaccines-14-00039-t002:** Overview of the serological IBV M41 and IBV 4–91 HI titers at time of different IBV challenges. Titers displayed as mean ± standard deviation. * = *p* < 0.0001 with exact Wilcoxon rank-sum test compared to non-vaccinated birds.

Challenge Strain	Age in Weeks at Time of Challenge	Group (Number of Birds)	IBV M41 Titer Before Challenge (log_2_ HI)	IBV 4–91 Titer Before Challenge (log_2_ HI)
IBV M41	26	Non-vac (40)	4.1 ± 0.3	4.8 ± 0.4
		Multriva only (40)	7.6 ± 1.3 *	8.4 ± 1.3 *
		Prime+boost (40)	10.0 ± 1.3 *	9.3 ± 1.1 *
	60	Non-vac (20)	4.9 ± 0.3	4.3 ± 0.4
		Prime+boost (20)	10.0 ± 0.7 *	11.1 ± 0.8 *
	96	Non-vac (20)	4.6 ± 0.9	4.9 ± 1.5
		Prime+boost (20)	9.8 ± 2.2 *	11.0 ± 1.2 *
IBV 4–91	27	Non-vac (48)	4.2 ± 0.4	4.3 ± 0.5
		Multriva only (48)	7.3 ± 1.3 *	9.5 ± 0.9 *
		Prime+boost (48)	8.0 ± 1.2 *	10.5 ± 1.0 *
	60	Non-vac (20)	4.9 ± 0.4	5.0 ± 0.2
		Prime+boost (20)	10.6 ± 1.7 *	11.3 ± 1.2 *
	96	Non-vac (20)	4.8 ± 0.6	4.7 ± 0.6
		Prime+boost (20)	10.4 ± 1.2 *	11.2 ± 3.3 *
IBV QX	40	Non-vac (26)	4.6 ± 0.5	5.0 ± 0.9
		Prime+boost (30)	9.7 ± 1.4 *	11.4 ± 1.0 *
IBV Q1	26	Non-vac (30)	4.6 ± 0.5	4.4 ± 0.5
		Prime+boost (30)	10.1 ± 1.1 *	12.0 ± 0.9 *
IBV Var2	37	Non-vac (30)	4.6 ± 0.5	4.6 ± 0.5
		Prime+boost (30)	9.7 ± 1.2 *	11.3 ± 0.9 *

**Table 3 vaccines-14-00039-t003:** Overview of the total score per group of coughing occurrences after each IBV challenge. IB-related clinical signs (coughing/snicking) were observed daily for 4 weeks post-challenge. Scores were given as coughing in a pen (score 1) or multiple birds coughing in a pen (score 2). The total score per group was calculated by adding all individual daily scores recorded in all pens of each group. The higher the total score, the higher/longer coughing occurrence per group.

Challenge Strain	Age in Weeks at Time of Challenge	Group	Total Score (Clinical Signs of Coughing)
IBV M41	26	Non-vac	31
		Multriva only	4
		Prime+boost	4
	60	Non-vac	18
		Prime+boost	0
	96	Non-vac	22
		Prime+boost	1
IBV 4–91	27	Non-vac	15
		Multriva only	13
		Prime+boost	0
	60	Non-vac	23
		Prime+boost	0
	96	Non-vac	20
		Prime+boost	0
IBV QX	40	Non-vac	36
		Prime+boost	25
IBV Var2	37	Non-vac	34
		Prime+boost	2

## Data Availability

All data supporting the findings of this study are included within the manuscript.

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
