# Peer review of "Inactivated Avian Infectious Bronchitis Virus Strains M41 and 4–91 Provide Broad Protection Against Multiple Avian Infectious Bronchitis Strains"

_vaccines, 2025, doi:10.3390/vaccines14010039_

Round 1

Reviewer 1 Report

Comments and Suggestions for Authors

The authors tested combinations of inactivated infectious bronchitis virus (IBV) vaccines commercially available to investigate their efficacies on IBV disease in SPF and farmed chickens. IBV disease in chickens is a significant problem for the poultry industry. Therefore, this research presents valuable information for stakeholders to safeguard their poultry farming operations. This reviewer has the following suggestions:  

Specific comments: 

Line 46: "it may result"

Line 47: Family name should be italicised, as per the ICTV convention. 

Lines 53-55: A recent review article reported the occurrence of nine IBV genotypes (https://pmc.ncbi.nlm.nih.gov/articles/PMC10875066/). This needs to be elaborated a little bit in terms of their geographical distribution. 

Line 63 and many other places: "vaccine regimen" is a more common and widely acceptable term than "vaccine regime". I leave it to the discretion of the authors to decide whether they still wish to maintain their choice of terminology. 

Lines 65-73: There is a lack of coherence in the flow of information. Please, before starting this paragraph, briefly mention the available commercialised IBV vaccines and the associated challenges with those vaccines, which prompted you to hypothesise this study. 

Line 329: Please define clearly what those nine antigens are. 

Lines 334-335: Please provide more specific information on the vaccine's efficacy. 

General comments: 1. Please provide a clear statement on the circulation and prevalence of M41 and 4-91 strains of IBV in different geographic locations. In which continents are these two IBV strains prevalent?

2. Please clearly describe in the Introduction section what this nine-valent vaccine is. It is not clear at the moment. Have similar studies been performed in the past? 

3. English can be corrected/improved at a few places. 

Comments on the Quality of English Language

Careful proofreading is required. 

Reviewer 2 Report

Comments and Suggestions for Authors

The authors have written this manuscript in accordance with the journal's requirements. Sufficient information is presented in all sections of the manuscript, so I don't have any recommendations for the authors.

Reviewer 3 Report

Comments and Suggestions for Authors

The manuscript titled “Inactivated avian infectious bronchitis virus strains M41 and 4-91 provide broad protection against multiple avian infectious bronchitis strains” by van de Weem et al. describes the use of novel inactivated IBV vaccines containing two of the IBV strains, M-41 and 4-91. The novel inactivated IBV vaccine contains nine different antigens and provides cross-protection against homologous and heterologous challenged IBV strains genotypes representing different geographical regions. Both vaccine regimes; prime-boost and Multriva, have provided high HI titer antibodies compared to the non-vaccinated group. Although the result is promising, the experimental methods would have been improved by including a viral shedding study and some histological data from the vaccinated challenge group, particularly for nephrogenic strains like QX and Var2 strains.

The protection study data was mainly based on HI titer and a reduction in the clinical signs score in the challenged group compared to the non-vaccinated group. Although there are several past vaccine studies where viral shedding, cilostasis score, and histopathological findings were included to study the vaccine efficacy.

The intended vaccine could be a promising candidate for IBV vaccine strategies in the layer chickens and will provide broader protection against different genotypes of IBV strains for an extended period of time.

I have the following comments regarding the manuscripts:

1.     How many birds (n=) were included in each study group? Nowhere is it mentioned.

2.     Table 1: In the prime and boost column, the age at which these birds were vaccinated could have been mentioned by providing the asterisk and explaining it in the footnote of the table.

3.     Nowhere in the experimental studies is the rationale for selecting time points for homologous challenge viruses (26 wks, 27 wks, 60 wks, and 96 wks) as well as heterologous viruses (26 wks, 37 wks, and 40 wks) mentioned.

4. Line 114 time points for monitoring egg production after challenge are 26, 60, and 96 woa, but in table 2 for IBV 4-91, the time points mentioned are 27, 60, and 96 woa. Is it a typographical error?

5.     Provide the reference for the HI methodology (subsection 2.7) and the source of the antigens used in the assay.

6.     Why is there no specific HI titer of <=5 log2, and whether it has been confirmed by any other assays like VN or ELISA that it's non-specific (table 2 and elsewhere)?

7.     Why was the vaccine challenge study result not provided beyond 26 woa for M-41 and 27 wks for 4-91 for the Multrivia groups (Table 2)?

8.      Why the Multrivia group was not further challenged at 60 woa and 96 woa for homologous challenge viruses and 26, 37, and 40 woa for non-homologous viruses.

9.     Line 246, is it reminded high  or remained high?

Reviewer 4 Report

Comments and Suggestions for Authors

Overall, the manuscript lacks information about the experiments used and is not described accurately, making it very difficult to understand. Also, the results of this experiment do not align well with the vaccination schedule of outdoor farms.

  1. A detailed description of the materials and methods is generally needed.
  2. Without information on the 9-valent inactivated vaccine, this manuscript would be very difficult to understand.
  3. Without information on the number of birds used in this study, it would be very difficult to give much meaning to the results presented in the figures.
  4. In line 200, the sentence “Several different commercial breeds of chickens (layers: Lohman Brown Lite and H&N Super Nick, broiler breeders: Ross 308) were vaccinated with a nine-valent inactivated vaccine between 12-15 woa as part of field vaccination protocols” has been described. There is no accurate data on the presence of IBV infection or vaccination in chickens raised on farms prior to vaccination. Therefore, the results analyzed after vaccination in 12-15-week-old chickens are considered to reflect an error in the experimental design.
  5. In farms, it is common to administer the IB vaccine before 1 week of age and the second vaccine at 3-4 weeks of age, but in this study, it is difficult to understand theoretically why the IB vaccine was administered at 15-16 weeks of age and the results were analyzed subsequently.
  6. In laying hens, IBV causes oviduct lesions such as hypertrophy, reducing egg production. Therefore, this study omits crucial histological observations.
  7. As shown in Figures 4 and 5, it is important to consider the possibility that IB challenge may result in reduced egg production due to histological problems in the reproductive tract of laying hens. Therefore, egg production should be continuously monitored for 4 to 10 weeks after challenge vaccination.
  8. The authors mainly observed clinical symptoms related to the respiratory system, but in laying hens, if egg production is reduced, analysis related to the reproductive system is considered more important.

Round 2

Reviewer 1 Report

Comments and Suggestions for Authors

I am satisfied with the revision and have no more suggestions. Well done! 

Author Response

Thanks for your comments.

Reviewer 3 Report

Comments and Suggestions for Authors

I have only one comments is to provide the full name of antigens included in vaccine , IBDV, ARV, NDV, AMPV and EDSV(Lines 78-79).

Author Response

The comment is as follows:
"I have only one comments is to provide the full name of antigens included in vaccine , IBDV, ARV, NDV, AMPV and EDSV (Lines 78-79)."
Response: Thank you for this comment. Nobilis Multriva contains the following nine antigens: two IBDV strains (GB02 and 89/03), two ARV strains (ARV-1 and ARV-4), one NDV strain (Ulster), one AMPV strain (BUT1 #8544), one EDSV strain (BC14), and two globally prevalent IBV strains (M41; genotype GI-1 and 4-91; genotype GI-13). To improve clarity, we have added this information to the revised manuscript in the Introduction, Materials and Methods, and Discussion sections (lines 78–82, 112–115, and 371–375).

Reviewer 4 Report

Comments and Suggestions for Authors

Much has been clarified, but additional information is still needed.
1. For further clarity, please add the inactivated vaccine (Nobilis multiva) to line 20 and elsewhere.
2. The antibody titers for multiva (dark yellow) at weeks 4–15 are shown in Figures 2 and 3. Do data on multiva's antibody titers at weeks 4–15 actually exist? No vaccination was administered during this period.

Author Response

1. For further clarity, please add the inactivated vaccine (Nobilis multiva) to line 20 and elsewhere.

Response 1: Thank you for your comment. We have incorporated the requested clarification by adding the inactivated vaccine name (Nobilis Multiva) to line 21 and throughout the manuscript where relevant. This adjustment ensures greater clarity and consistency in the text.

2. The antibody titers for multiva (dark yellow) at weeks 4–15 are shown in Figures 2 and 3. Do data on multiva's antibody titers at weeks 4–15 actually exist? No vaccination was administered during this period.

Response 2: Thank you for your observation.

Regarding Figure 2: The dataset for Multriva (dark yellow) originates from SPF animals. Birds in this group were not vaccinated with any product prior to week 15, when they received the inactivated vaccine (Nobilis Multriva). The data shown for weeks 4–15 represent baseline serology, confirming that these birds remained seronegative throughout this period. Antibody titers observed after week 15 are therefore exclusively attributable to Multriva vaccination. As detailed in lines 131–133 and 198–200, blood samples were collected at weeks 1, 4, 8, and 12 and analyzed using IB HI tests. Vaccination with Multriva was performed at week 15.

Regarding Figure 3: The dataset for H&N Super Nick Layers (dark yellow) reflects commercial field animals that were primed with IB live vaccines, as presented in Table 2 and described in Section 2.5 (lines 143–147). Consequently, the IB HI titers observed at weeks 4–15 in these animals result from the live IB priming vaccinations.